# Quality of endodontic record-keeping and root canal obturation performed by final year undergraduate dental students: An audit during the COVID-19 pandemic

**Galvin Sim Siang Lin**[1]*, **Wen Wu Tan**[2], **Daryl Zhun Kit Chan**[1], **Kah Hoay Chua**[3☯], **Teoh Chai Yee**[3☯], **Mohd Aizuddin Mohd Lazaldin**[4☯]

1 Department of Dental Materials, Faculty of Dentistry, Asian Institute of Medicine, Science and Technology (AIMST) University, Bedong, Kedah, Malaysia, 2 Department of Dental Public Health, Faculty of Dentistry, Asian Institute of Medicine, Science and Technology (AIMST) University, Bedong, Kedah, Malaysia, 3 Department of Dental Technology, Faculty of Dentistry, Asian Institute of Medicine, Science and Technology (AIMST) University, Bedong, Kedah, Malaysia, 4 Department of Biosciences, Faculty of Science, Univeristi Teknologi Malaysia, Johor, Malaysia

☯ These authors contributed equally to this work.
* galvin@aimst.edu.my

**Data Availability Statement:** Data cannot be shared publicly due to the potential risk of

## Abstract

### Background

To assess the quality of endodontic record-keeping and root canal obturation performed by undergraduate final year dental students during the COVID-19 pandemic.

### Materials and methods

Dental records and dental radiographs of patients who received endodontic treatments between March 2020 and March 2022 by undergraduate students during the COVID-19 pandemic were included. The audit data were collected based on the European Society of Endodontology guidelines retrospectively via assessing the patient's clinical records and intraoral periapical radiograph. Root canal obturation quality was evaluated based on the following parameters: adaptation, length, taper, and mishap. A root filling is defined as satisfactory only when all four parameters were graded as acceptable. Subsequently, the data were recorded and analysed using Chi-Square test with the level of significance set at p = 0.05.

### Results

A total of 111 patient records with 111 root canal-treated teeth were evaluated. The highest percentage of documented evidence was noted in the patient's general records, while the endodontic treatment records showed the lowest percentage of documented evidence. 78 (70.3%) of root canal-treated teeth were deemed satisfactory with acceptable adaptation, length, taper, and absence of mishap. Moreover, no statistical significance in terms of root canal filling quality was noted between anterior and posterior teeth, and between maxillary and mandibular arch.

disclosing patient's personal information. Data will be available upon request from the AIMST University Human & Animal Ethics Committee (contact via +6044298000 or aimstrmc@aimst. edu.my) for researchers who meet the criteria for access to confidential patient data.

**Funding:** The authors received no specific funding for this work.

**Competing interests:** The authors have declared that no competing interests exist.

## Conclusions

Although patient records and root canal fillings quality were deemed satisfactory in most cases, strict documentation requirements and continuing dental education in audit training are necessary for quality assurance.

## Introduction

Maintaining proper patient records is an important aspect of daily dental practice as the delivery of high-quality patient care that complies with professional and legal requirements depends on documentation [1]. Every dental practitioner is responsible for keeping up-to-date, comprehensive, clear, precise, and readable dental records [2]. A complete history of patient management can be formulated with well-documented records, ensuring an effective and orderly flow of treatment planning [3]. Moreover, high-quality dental records are also needed for intra- and inter-clinician communication as well as dental-legal documentation [4]. Although good clinical records may not guarantee satisfactory dental care, they do facilitate effective and thorough quality assurance evaluation. Thus, dental institutes serve a significant role in the development of future dental practitioners' record-keeping competence.

Undeniably, undergraduate dental students are frequently required to provide treatment to a single patient over several appointments, including endodontic treatment. Due to the enormous amount of information that needs to be documented, keeping patient records for endodontic treatment can be challenging [2]. During endodontic procedures, a wide range of endodontic materials, stainless steel hand files, nickel-titanium rotary file systems, and obturation techniques are employed, all of which should be accurately recorded in the patient notes [5]. Since endodontic treatment is not always accomplished in a single visit, proper documentation enables dental students and dental practitioners to quickly review earlier records and resume treatment without having to repeat or omit treatment phases. Furthermore, extensive records are invaluable to monitor the long-term endodontic outcomes [2].

In 2006, the European Society of Endodontology (ESE) released the 'Quality Guidelines for Endodontic Treatment' consensus report, which aimed to represent current excellent practice and thereby standardise endodontic treatment quality [6]. According to ESE guidelines, endodontic treatment is also a necessary academic requirement for dental students, and a well-obturated root canal should be free of void, with the root filling material placed within 0.5–2 mm of the radiographical apex to prevent post-treatment complications [6]. Therefore, it is necessary to evaluate dental students' endodontic treatment performance in order to ensure patient safety and a high-quality curriculum [7]. Several root canal filling audits have been conducted among dental students by examining radiographs taken during clinical practice and comparing them to pre-determined benchmarks [5, 8–10]. However, the available findings revealed mixed results on the acceptability of root canal obturation performed by dental students ranging from 36% to 80.2% [8–11]. Nonetheless, these studies were conducted before the outbreak of the severe acute respiratory syndrome coronavirus 2 (SARS-CoV-2), known as COVID-19.

COVID-19 has wreaked havoc on dental education around the world, with the majority of dental institutions and universities shutting down as a result of the pandemic, leading to limited clinical training among undergraduate students [12]. In addition, strict measurements were implemented to reduce the amount of time and frequency of student-patient contact due to the close proximity of dental students and patients when performing dental procedures

[13]. The authors speculated that the pandemic's restriction on clinical sessions and patient treatment visits would have an impact on students' record-keeping performance. Thus, evaluating the students' performance during this period is critical, as it will provide valuable information to the current dental education in ensuring the learning outcomes and competencies are achieved. It is also crucial to underline the need of establishing policies that support the monitoring of patient documentation and the value of record-keeping in dental practice. To the best of the authors' knowledge, the present study is the first of its kind to assess the quality of endodontic record-keeping and root canal obturation performed by undergraduate final year dental students in a Malaysian dental institution during the COVID-19 pandemic.

## Materials and methods

### Sample size calculation

Sample size was calculated based on a previous similar study using G*Power 3.1 software [9]. The proportions of acceptability of root canal obturation based on the Fong *et al.* [9] were 0.72 and 0.91 for length and adaptation quality, respectively. A z-test was employed with the alpha = 0.05, power = 0.8, and allocation of ratio = 1, resulting in a total sample size of 102.

### Study selection

Ethical clearance was granted by the Asian Institute of Medicine, Science and Technology (AIMST) University Human & Animal Ethics Committee with the ethical approval code: AUHEC/FOD/2022/02. No further consent from the patient is needed since written consent forms for patient folder confidentiality has been signed before receiving any dental care. A retrospective clinical audit was conducted at the Faculty of Dentistry, AIMST University Malaysia in which all dental records and dental radiographs of patients who had given consent and received endodontic treatments by undergraduate dental students during the COVID-19 pandemic were included in the current study. In this context, the COVID-19 pandemic began in March 2020 and ended in March 2022, as the country entered the endemic phase in April 2022. Hence, the study includes only patients treated from March 2020 to March 2022.

### Endodontic record-keeping

The audit data were collected based on the ESE guidelines by two investigators (senior faculty members) retrospectively via assessing the patients' clinical records [6]. Endodontic record-keeping was evaluated under the following headings: general records, radiographic records, and endodontics treatment records. Each component was documented as: recorded correctly (Y), not recorded (N) and not applicable to the patient's treatment (NA). All criteria were recorded as a single entry of data in a customized google spreadsheet form. In addition, NA scores were excluded from calculations, and Y and N scores were used to calculate percentages.

### Root canal obturation quality

Intraoral periapical radiographs were retrieved from each included patient record. The inclusion criteria are: (1). Single or multi-rooted permanent tooth, (2). Treatment undertaken by final year undergraduate dental students under the supervision of clinical lecturers, (3). Treatment undertaken between March 2020 and March 2022, (4). Presence of a post-obturation radiograph showing the entire root length, (5). At least 2–3 mm of the periapical area beyond the root apex. On the other hand, the exclusion criteria are: (1). Post-obturation radiographs

were not available, (2). Post-obturation radiographs showed artefact or superimposition, (3). Treatments undertaken before COVID-19 outbreak, (4). Retreatment of root canal.

As a gold standard for the technical quality of the root fillings, the (ESE) quality guideline for endodontic treatment was adopted [6], and the criteria to assess root canal obturation are listed in Table 1. In general, the radiographs were evaluated based on the following parameters: adaptation, length, taper, and mishap. Either 'acceptable' or 'unacceptable' was graded for adaptation, length, and taper. Meanwhile, mishap of root canal filling was evaluated as 'presence' or 'absence'. A root filling is defined as satisfactory only when all parameters were graded as acceptable with the absence of mishap [9]. Prior to the start of the assessment, two investigators (senior faculty members) were trained in using the assessment criteria and were calibrated based on the Cohen's Kappa for intra- and inter-examiner reliability and reproducibility of results. A third investigator was referred to in case of uncertainty.

## Data analysis

Subsequently, the data were recorded in a Google Excel spreadsheet and later transferred to SPSS software version 26.0 for statistical analysis. Descriptive data were recorded as percentage and frequencies while the comparisons of treatment outcomes between groups (anterior and posterior teeth; maxillary and mandibular teeth) were calculated using Chi-Square test with the level of significance set at $p = 0.05$. Intra- and inter-investigator reliability and reproducibility were assessed by Cohen's Kappa statistics. Each investigator was asked to assess 15 radiographs twice for intra-examiner reliability, and the result was compared with the second investigator for inter-examiner reliability. The *K*-values were acceptable for all the parameters ($k > 0.8$).

# Results

## Endodontic record-keeping

A total of 111 patient records were retrieved and evaluated in the present study. The audit outcomes are listed in Table 2. Among the 111 patient records, only 70 (63%) showed evidence of post-obturation radiograph. Hence, only endodontically treated teeth of these 70 patients were included in the root canal obturation quality audit. Among the 70 patients, 111 teeth that had undergone endodontic treatment were retrieved. Male patients comprised 38.7% (n = 43) of

**Table 1. Criteria used to assess the quality of root canal obturation.** Adopted from Wong CY *et al.* [10].

| Parameter | Criteria | Description |
|---|---|---|
| Adaptation | Acceptable | • No void identified in root canal filling or between root canal filling and root canal walls |
| | Unacceptable | • Presence of void in root canal filling or between root canal filling and root canal walls |
| Length | Acceptable | • Root canal filling material is within the root canal system and ending 0–2mm of the radiographic apex |
| | Unacceptable | • Root canal filling material is >2mm short of the radiographic apex (Under-filled).<br>• Root canal filling material is extruded beyond the radiographic apex (Over-filled) |
| Taper | Acceptable | • Consistent taper from the orifice to the apex |
| | Unacceptable | • No consistent taper from the orifice to the apex |
| Mishaps | Absent | • No mishap identified |
| | Present | • Root canal filling is at least 1 mm shorter than the working length and is deviated from the original canal curvature (ledge).<br>• Apical termination of filled canal is different from the original canal terminus or root canal filling material is extruded through the apical foramen (perforation).<br>• Separation of instruments |

**Table 2. Documentation of patient general, radiographic, and endodontic treatment records.**

|  | Yes, n (%) | No, n (%) |
|---|---|---|
| **General Records** | | |
| Presenting Symptoms | 107 (96.4) | 4 (3.6) |
| History of Presenting Complaint | 101 (91) | 10 (9) |
| Clinical Examination | 111 (100) | 0 (0) |
| Sensibility Test | 94 (84.7) | 17 (15.3) |
| Diagnosis | 108 (97.3) | 3 (2.7) |
| Treatment Plan | 108 (97.3) | 3 (2.7) |
| Consent | 111 (100) | 0 (0) |
| **Radiographic Records** | | |
| Pre-operative Radiograph | 104 (93.7) | 7 (6.3) |
| Working Length Radiograph | 84 (75.7) | 27 (24.3) |
| Master Gutta-percha Radiograph | 100 (90) | 11 (10) |
| Post-obturation Radiograph | 70 (63.1) | 41 (36.9) |
| **Endodontics Treatment Records** | | |
| Use of Local Anaesthesia | 62 (55.9) | 49 (44.1) |
| Rubber Dam Isolation | 54 (48.6) | 57 (51.4) |
| Notable findings | 17 (100)* | NA |
| Working Length & Reference Point | 110 (99.1) | 1 (0.9) |
| Size of Canal Preparation | 105 (94.6) | 6 (5.6) |
| Preparation Technique | 42 (37.8) | 69 (62.2) |
| Volume and concentration of Irrigant | 44 (39.6) | 67 (60.4) |
| Intracanal Dressing | 85 (76.6) | 26 (23.4) |
| Temporary Restoration | 104 (93.7) | 7 (6.3) |
| Medication Prescribed | 5 (100)* | NA |
| Filling Materials | 57 (51.4) | 54 (48.6) |
| Complication | 5 (100)* | NA |
| Advice on final restoration | 88 (79.3) | 23 (20.7) |
| Outcome | 16 (85.6) | 95 (14.4) |

NA: Not applicable

*Applicable to specific cases

the sample, while female patients comprised 61.3% (n = 68). The highest percentage of documented evidence was noted in the patient's general records with all the components scoring more than 84%, while the endodontic treatment records showed the lowest percentage of documented evidence with rubber dam isolation, preparation technique and volume and concentration of irrigant were recorded less than 50% of the cases.

## Root canal obturation quality

Table 3 shows that a total of 111 teeth were assessed for the quality of obturation, including 39 (35.1%) anterior teeth and 72 (64.9%) posterior teeth. 68.5% (n = 76) were maxillary teeth while 31.5% (n = 35) were mandibular teeth. The present study included 29.7% (n = 33) incisors, 5.4% (n = 6) canine, 28.8% (n = 32) premolars and 36.1% (n = 40) molars. Moreover, the obturation quality in anterior teeth showed that 87.2% (n = 34) of the canal adaptation, 74.4% (n = 29) of the length, and 97.4% (n = 38) of the taper were considered acceptable, respectively, with no canal mishap noted (Table 3). In contrast, the adaptation and taper qualities of posterior teeth were slightly lower (79.2% and 93.1%, respectively) with a higher occurrence of canal

**Table 3. Quality of root canal fillings according to tooth location and arch type.**

| Variables | Adaptation | | Length | | Taper | | Mishap | |
|---|---|---|---|---|---|---|---|---|
| | Acceptable | Unacceptable | Acceptable | Unacceptable | Acceptable | Unacceptable | Absent | Present |
| **Tooth Location** | | | | | | | | |
| Anterior (n = 39) | 34 (87.2%) | 5 (12.8%) | 29 (74.4%) | 10 (25.6%) | 38 (97.4%) | 1 (2.6%) | 39 (100%) | 0 (0%) |
| Posterior (n = 72) | 57 (79.2%) | 15 (20.8%) | 54 (75%) | 18 (25%) | 67 (93.1%) | 5 (6.9%) | 68 (94.4%) | 4 (5.6%) |
| | p = 0.294 | | p = 0.941 | | p = 0.663 | | p = 0.295 | |
| **Arch Type** | | | | | | | | |
| Maxillary (n = 76) | 62 (81.2%) | 14 (18.8%) | 59 (77.6%) | 17 (22.4%) | 72 (94.7%) | 4 (5.3%) | 74 (97.4%) | 2 (2.6%) |
| Mandibular (n = 35) | 29 (82.9%) | 6 (17.1%) | 24 (68.6%) | 11 (31.4%) | 33 (94.3%) | 2 (5.7%) | 33 (94.3%) | 2 (5.7%) |
| | p = 0.871 | | p = 0.307 | | p = 0.988 | | p = 0.589 | |

mishaps (5.6%). Nevertheless, a comparison of obturation quality between anterior teeth and posterior teeth showed no significant difference (p > 0.05). Although maxillary teeth showed satisfactory obturation quality in terms of length (81.2%, n = 59) and taper (94.7%, n = 72) as compared to mandibular teeth, no significant difference (p > 0.05) was noted. In general, 78 (70.3%) root canal-treated teeth were deemed satisfactory.

## Discussion

Undergraduate dental curriculum has been acknowledged as unique to the profession due to the substantial amount of hands-on training included in their curriculum [14]. It is noteworthy to mention that the standard of training received by these undergraduate students will likely have a ripple effect on the calibre of their future dental practices. With the emergence of COVID-19, the globe has brought about a paradigm shift and the practice of dentistry has been significantly altered [15]. However, the need to adhere to the professional requirements for record-keeping should not be affected or diminished by enduring a global pandemic and handling the constraints on dental treatment. Dental students and dental practitioners must continue to keep accurate records of all patient treatments to reflect how local regulations may have influenced patients' experiences during the pandemic. Thus, conducting a routine clinical audit is a necessity in some dental institutions to improve patient oral care and service delivery, provide an opportunity for audit training, and identify areas that require improvement in the oral health care delivery system [16].

The current results revealed that documentation of patient's consent and clinical examination were found in all cases which contradicts previous studies [2, 17]. Informed consent must be obtained before patients make any decision and before any dental procedure as it serves as the cornerstone of building trust between the patient and clinician [17]. Although most of the items were identified in the patient's general records with more than 90% compliance, evidence of sensibility test was still found to be missing in several records. Undeniably, dental students and dental practitioners can benefit from pulp sensibility tests such as cold test and electric pulp test, since these tests offer useful information for diagnosis and treatment planning [18]. If pathosis is evident, pulp testing in conjunction with information from the patient's history, clinical and radiographical examinations will lead to the provisional diagnosis of the underlying pulpal and periapical diseases. Even though a significant drawback of these tests is that they only inadvertently reveal the status of the pulp by assessing neural response rather than vascular supply [19], it is still essential to perform and document pulp sensibility test findings in the patient's dental record which can aid in endodontic diagnosis and dent-legal issues.

Among the radiographic records, post-obturation radiographs exhibited the least compliance, with only 63.1% being documented, which is lower than that of a previous study [20]. Obturation is an endodontic procedure used to create a fluid-tight barrier that shields the periapical tissues from oral cavity-dwelling microorganisms [21]. Although the effectiveness of the root canal obturation cannot be completely assessed through radiograph, the root canal filling adaptability, length, and taper are often assessed based on the post-obturation radiograph. One possible explanation for the low compliance in documenting post-obturation radiograph could be due to the time restriction on clinical sessions imposed by the dental school, causing students to rush their clinical tasks within a short period. Another plausible reason could be the poor knowledge among students, wherein students might believe that the master gutta-percha radiograph suffices to foretell the success of obturation quality. Additionally, the COVID-19 pandemic had a negative influence on dental education, making it more difficult for teachers to adequately supervise their students' work in the clinics [22]. Nonetheless, future studies are required to explore students' learning perception of endodontic subjects during the COVID-19 pandemic to establish a correlation with their endodontic record-keeping performance.

Furthermore, the use of rubber dam, preparation technique, volume and concentration of irrigating solution are substantially underreported (<50% compliance) in the current study which corroborates other studies [2, 20]. This can be explained by the range of information needed for thorough treatment records makes it difficult to maintain high-quality endodontic record keeping, as recommended by the European Society of Endodontology [6]. The use of endodontic record-keeping templates in conjunction with record-keeping training is one strategy to combat this and encourage dental students and clinicians to document all pertinent treatment information, which has been found to enhance endodontic record-keeping compliance [2].

The present results indicated that mishaps were the parameter that occurred with the lowest frequency, while acceptability rates of root canal taper, adaptation, and length were generally high (> 65%). Nevertheless, the present findings contradict a previous study suggesting that only 13.1% of endodontically treated teeth performed by dental students had adequate adaptation and 14.2% had adequate taper [5]. The disparity in the results may be driven by the slight variations in the assessment criteria and the fact that the current study only included undergraduate dental students in their final year. It has been suggested that final year dental students are more confident in performing endodontic treatment and that students with greater experience in performing such cases would demonstrate a higher self-efficacy than junior learners [23–25]. Nevertheless, a direct comparison might not be reasonable considering that the previous study was conducted on extracted teeth [5], whereas the current study involved retrospective evaluation of real patients.

The present study showed that endodontically treated anterior teeth had slightly higher acceptability in root canal adaptation, length, and taper with lower frequency of mishap than posterior teeth which is in line with previous similar studies [5, 26, 27]. This could be explained by the challenging treatment procedure in posterior teeth with multiple root canals and complex root canal anatomy, whilst anterior teeth mostly have single canal which is more straightforward [28, 29]. Another rationale behind this is that the Malaysian Qualification Agency (MQA) established the required minimal clinical experience (MCE) and expected clinical experience (ECE) of completing root canal treatment on three anterior teeth and only one posterior tooth prior to graduation [23]. The global outbreak of the COVID-19 pandemic has also rendered the clinical exposure to be reduced and it has been reported that finding patients who need endodontic treatment, particularly for molar cases, is challenging during the pandemic [23]. Therefore, it is not surprising that dental students demonstrated lower satisfaction

with managing posterior teeth, particularly molars, when they had limited exposure to it. In the current study, maxillary teeth generally showed greater acceptance in root canal obturation outcomes as compared to mandibular teeth, albeit with no significant difference, which is in accordance with Elemam RF *et al.* [27] and Ammar A *et al.* [11], but contradicts that of Elsayed RO *et al.* [30].

One of the limitations was that the factors influencing patient's documentation were not examined in the current study. Additionally, methodological flaws could have been introduced due to the inherent limits of radiographic evaluation and interpretation, as the radiographic appearance may be impacted by changes in beam and film angulation [5]. The overall quality of endodontic treatment could be influenced by the type of instrumentation and obturation technique as most undergraduate training employ conventional stainless-steel hand files with step-back and cold lateral condensation techniques [11, 30]. Nevertheless, stringent documentation standards and continuing dental education are warranted to guarantee that dental students adhere to ethical conduct and are conversant with institutional documentation policies. One way to improve this is to provide dental students with a dental documentation template to record dental charts and establish consistency in dental record-keeping among the students. Clinical audit may be seen as a valuable teaching tool that helps improve students' understanding of the importance of record-keeping and should be included in the undergraduate dental curriculum [16]. Additionally, it should be considered to establish a record-keeping policy for the teaching staff and conduct regular audits to ensure record-keeping standards are followed and maintained [2]. The present study also offers baseline data for dental practitioners in the public or private healthcare system to monitor their dental records as part of the quality assurance protocol.

## Conclusion

Within the constraints of the present findings, the patient's general and radiographic records were well-documented except for evidence of post-obturation radiograph. Meanwhile, the evidence of endodontic treatment record was well-documented except for evidence of rubber dam isolation, preparation technique and volume and concentration of irrigant. Root canal fillings performed by undergraduate students during the COVID-19 pandemic were deemed satisfactory in 70.3% of the cases with no significant difference noted in terms of tooth location and arch type. Nonetheless, strict documentation requirements and continuing dental education in audit training are necessary as patient safety and treatment quality are major considerations in the provision of healthcare. Future research should investigate how clinical audit is utilised in various clinical settings and its effectiveness for quality assurance.

## Author Contributions

**Conceptualization:** Galvin Sim Siang Lin.

**Data curation:** Galvin Sim Siang Lin, Wen Wu Tan.

**Formal analysis:** Galvin Sim Siang Lin.

**Investigation:** Galvin Sim Siang Lin, Wen Wu Tan, Daryl Zhun Kit Chan, Kah Hoay Chua, Teoh Chai Yee, Mohd Aizuddin Mohd Lazaldin.

**Methodology:** Galvin Sim Siang Lin, Wen Wu Tan, Daryl Zhun Kit Chan, Kah Hoay Chua, Teoh Chai Yee.

**Project administration:** Galvin Sim Siang Lin.

**Resources:** Galvin Sim Siang Lin, Wen Wu Tan.

**Software:** Galvin Sim Siang Lin, Wen Wu Tan, Mohd Aizuddin Mohd Lazaldin.

**Validation:** Wen Wu Tan.

**Writing – original draft:** Galvin Sim Siang Lin, Wen Wu Tan.

**Writing – review & editing:** Daryl Zhun Kit Chan, Kah Hoay Chua, Teoh Chai Yee, Mohd Aizuddin Mohd Lazaldin.

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
