## [Decision Letter · Decision Letter 0]

3 Aug 2022

PONE-D-22-17740Quality of Endodontic Record-keeping and Root Canal Obturation Performed by Final Year Undergraduate Dental Students: An Audit During the COVID-19 PandemicPLOS ONE

Dear Dr. Lin,

Thank you for submitting your manuscript to PLOS ONE. After careful consideration, we feel that it has merit but does not fully meet PLOS ONE’s publication criteria as it currently stands. Therefore, we invite you to submit a revised version of the manuscript that addresses the points raised during the review process.

We look forward to receiving your revised manuscript.

Kind regards,

Ove A. Peters, DMD MS PhD

Academic Editor

PLOS ONE

Journal Requirements:

Additional Editor Comments:

The ms was reviewed by two content experts and requires minor changes as outlined in the reviewers' reports.

Reviewers' comments:

Reviewer's Responses to Questions

**Comments to the Author**

1. Is the manuscript technically sound, and do the data support the conclusions?

Reviewer #1: Yes

Reviewer #2: Yes

2. Has the statistical analysis been performed appropriately and rigorously? 

Reviewer #1: Yes

Reviewer #2: Yes

3. Have the authors made all data underlying the findings in their manuscript fully available?

Reviewer #1: Yes

Reviewer #2: Yes

4. Is the manuscript presented in an intelligible fashion and written in standard English?

Reviewer #1: Yes

Reviewer #2: Yes

5. Review Comments to the Author

Reviewer #1: The manuscript is structured, written in a concise manner and in standard English. The topic is interesting, sample size calculation is included in the study, the statistical analysis is appropriate and the results are presented with clarity. I would like to make the following two comments: a) there are no specifics about the status of the two investigators that assessed the PAs. It is important to know if they were undergraduate students, postgraduate students or faculty members. b) It would be preferable to include the p values in the manuscript.

Reviewer #2: Thank you for submitting your manuscript to PLOS One for possible publication. Below please find my reviewer comments.

Introduction

Please add and explanation on why you assume that record keeping, and treatment quality would be different during the Covid-19 Pandemic than during other periods, as you have defined this as your investigated timeframe.

Materials and Methods

Endodontic record keeping: How were the 111 patient records selected from all available records from March 2020 to March 2022? E.g., Randomly selected or the first 111? Certain number per year? Please explain.

Please check your tenses throughout the document, it should be past tense, e.g., NA score were excluded instead of will be excluded.

Results

Table 2 Legend: Instead of distribution this should be called documentation of records.

You state that the only acceptable mishaps were no mishaps.

Consequently, your results would be easier to read if you dichotomise into mishaps/no mishaps, instead of acceptable/unacceptable mishaps which could imply that there are such mishaps that are deemed acceptable.

This is also true for Table 3 and the relevant section under “root canal obturation quality”. You state here that the “mishap quality” was “slightly lower” in posterior teeth. Please correct this statement.

Discussion

Again, you state that “root canal mishaps had the highest acceptability rate”. This conveys the message that mishaps can be acceptable. This should be reworded, for example that mishaps were the variable that occurred with the lowest frequency.

Please discuss why there was such low compliance with post-obturation radiographs, and this is obviously of great importance to ensure that patients don’t leave the clinic before any treatment accidents such as overfilling into the IA canal are ruled out.

Looking forward, did you consider providing students with a documentation template that they could copy/paste into their patient chart to create consistency in record keeping?

Conclusion

You introduce the category “moderately” satisfactory, while stating in M&M that only if all 4 parameters were graded as acceptable, then the overall case would be satisfactory. Please explain and introduce into M&M or change accordingly.

6. PLOS authors have the option to publish the peer review history of their article (what does this mean?). If published, this will include your full peer review and any attached files.

Reviewer #1: No

Reviewer #2: No

---

## [Author Response · Author response to Decision Letter 0]

31 Aug 2022

Editor

Respond: The style requirements were checked.

2. Please provide additional details regarding participant consent. In the ethics statement in the Methods and online submission information, please ensure that you have specified what type you obtained (for instance, written or verbal, and if verbal, how it was documented and witnessed). If your study included minors, state whether you obtained consent from parents or guardians. If the need for consent was waived by the ethics committee, please include this information 

Respond: The phrase has been added:

“No further consent from the patient is needed since written consent forms for patient folder confidentiality have been signed before receiving any dental care.”

3. In your Data Availability statement, you have not specified where the minimal data set underlying the results described in your manuscript can be found. 

Respond: The data availability statement has been addressed.

4. Please review your reference list to ensure that it is complete and correct. 

Respond: The reference list has been amended.

Reviewer 1

1. The manuscript is structured, written in a concise manner and in standard English. The topic is interesting, sample size calculation is included in the study, the statistical analysis is appropriate and the results are presented with clarity. I would like to make the following two comments: a) there are no specifics about the status of the two investigators that assessed the PAs. It is important to know if they were undergraduate students, postgraduate students or faculty members. b) It would be preferable to include the p values in the manuscript. 

Respond: The authors would like to thank the reviewer for the constructive comments.

(a): The words “senior faculty members” have been added.

(b). p values were added to Table 3.

Reviewer 2

1. Introduction

Please add and explanation on why you assume that record keeping, and treatment quality would be different during the Covid-19 Pandemic than during other periods, as you have defined this as your investigated timeframe. 

Respond: The authors would like to thank the reviewer for the constructive feedback in improving the manuscript. 

Several points were added to the introduction:

“In addition, strict measurements were implemented to reduce the amount of time and frequency of student-patient contact …… have an impact on students' record-keeping performance.”

2. Materials and Methods

Endodontic record keeping: How were the 111 patient records selected from all available records from March 2020 to March 2022? E.g., Randomly selected or the first 111? Certain number per year? Please explain.

Please check your tenses throughout the document, it should be past tense, e.g., NA score were excluded instead of will be excluded. 

Respond: All dental records and dental radiographs of patients who gave their consent and received endodontic treatments from March 2020 to March 2022 were retrieved which has been addressed in the manuscript. 

No specific sampling method was applied since all records were used in the study.

The manuscript has been revised and changed the phrases accordingly to past tense.

3. Results

Table 2 Legend: Instead of distribution this should be called documentation of records.

You state that the only acceptable mishaps were no mishaps.

Consequently, your results would be easier to read if you dichotomise into mishaps/no mishaps, instead of acceptable/unacceptable mishaps which could imply that there are such mishaps that are deemed acceptable.

This is also true for Table 3 and the relevant section under “root canal obturation quality”. You state here that the “mishap quality” was “slightly lower” in posterior teeth. Please correct this statement. 

Respond: The word ‘distribution’ has been changed to ‘documentation’.

The sentences have been revised:

“Either ‘acceptable’ or ‘unacceptable’ was graded for adaptation, length, and taper. Meanwhile, mishap of root canal filling was evaluated as ‘present’ or ‘absent’. A root filling is defined as satisfactory only when all parameters were graded as acceptable with the absence of mishap.”

Table 3 has been modified.

The statement has been corrected in the result section.

4. Discussion

Again, you state that “root canal mishaps had the highest acceptability rate”. This conveys the message that mishaps can be acceptable. This should be reworded, for example that mishaps were the variable that occurred with the lowest frequency.

Please discuss why there was such low compliance with post-obturation radiographs, and this is obviously of great importance to ensure that patients don’t leave the clinic before any treatment accidents such as overfilling into the IA canal are ruled out.

Looking forward, did you consider providing students with a documentation template that they could copy/paste into their patient chart to create consistency in record keeping? 

Respond: The sentence has been rephased. 

“The present results indicated that mishaps were the parameter that occurred with the lowest frequency, while acceptability rates of root canal taper, adaptation, and length were generally high (> 65%).”

The reason for low compliance is discussed in the text:

“One explanation for this could be the time restriction on clinical sessions imposed by the dental school…during the COVID-19 pandemic to establish a correlation with their endodontic record-keeping performance.”

The sentence has been added:

“One way to improve this is to provide dental students with a dental documentation template to record dental charts and establish a consistency in dental record-keeping among the students.”

5. Conclusion

You introduce the category “moderately” satisfactory, while stating in M&M that only if all 4 parameters were graded as acceptable, then the overall case would be satisfactory. Please explain and introduce into M&M or change accordingly. 

Respond: The conclusion has been rephrased accordingly.

---

## [Editor Report · Decision Letter 1]

20 Sep 2022

Quality of Endodontic Record-keeping and Root Canal Obturation Performed by Final Year Undergraduate Dental Students: An Audit During the COVID-19 Pandemic

PONE-D-22-17740R1

Dear Dr. Lin,

We’re pleased to inform you that your manuscript has been judged scientifically suitable for publication and will be formally accepted for publication once it meets all outstanding technical requirements.

Kind regards,

Ove A. Peters, DMD MS PhD

Academic Editor

PLOS ONE

Additional Editor Comments (optional):

The authors have satisfactorily addressed all comments, the ms is now acceptable.

---

## [Editor Report · Acceptance letter]

26 Sep 2022

PONE-D-22-17740R1 

Quality of Endodontic Record-keeping and Root Canal Obturation Performed by Final Year Undergraduate Dental Students: An Audit During the COVID-19 Pandemic 

Dear Dr. Lin:

I'm pleased to inform you that your manuscript has been deemed suitable for publication in PLOS ONE. Congratulations! Your manuscript is now with our production department. 

Kind regards, 

on behalf of

Dr. Ove A. Peters 

Academic Editor

PLOS ONE